# A Scoping Review of the Recent Clinical Practice Regarding the Evaluation of Bone Mineral Density in Children and Adolescents with Neuromuscular Diseases

**DOI:** 10.3390/medicina59020312

**Published:** 2023-02-08

**Authors:** Georgia Antoniou, Panagiotis Masouros, Dimitrios V. Papadopoulos, Konstantinos C. Soultanis, Panagiotis Krallis, George Babis, Vasileios S. Nikolaou

**Affiliations:** 1Department of Orthopedic Surgery, “Nicosia” General Hospital, 2029 Nicosia, Cyprus; 2Department of Orthopedic Surgery and Traumatology, “Evaggelismos” General Hospital, 10676 Athens, Greece; 32nd Department of Orthopaedic Surgery, “Konstantopouleio” General Hospital, National and Kapodistrian University of Athens, 14233 Athens, Greece; 41st Department of Orthopaedics, School of Medicine, National & Kapodistrian University of Athens, 12462 Athens, Greece; 52nd Department of Orthopedic Surgery, “Agia Sofia” General Children’s Hospital, 11527 Athens, Greece

**Keywords:** neuromuscular diseases, Duchenne muscular dystrophy, bone mineral density, fracture risk, evaluation

## Abstract

*Introduction:* Neuromuscular Diseases (NMD) are associated with decreased bone strength due to altered muscle–bone interaction. However, the evaluation of bone quality remains a certain challenge in these patients. The purpose of this scoping review is to investigate the recent literature regarding the assessment of Bone Mineral Density (BMD) in this population. *Methods:* An electronic search of the PubMed and Scopus database was performed considering studies published in the English literature after 2007 that evaluated BMD in pediatric and adolescent patients with NMD. We excluded studies that evaluated patients > 20 years, studies not involving humans, and studies investigating bone mineral density in various pediatric conditions, but without specific data on NMD. *Results:* Overall, 19 studies were included that evaluated BMD in 1983 patients with NMD. Duchenne Muscular Dystrophy was the most widely studied disease (n = 11 studies). Dual energy X-ray absorptiometry (DEXA) was the most common diagnostic modality for BMD evaluation, while the most frequent site for BMD measurement was the lumbar spine (89.4%, n = 17 studies), followed by total body BMD (68.4%, n = 13 studies). Low BMD in children with NMD was demonstrated in all studies, especially after loss of ambulation. Moreover, a positive correlation between lower BMD and older age was shown. *Conclusions:* BMD evaluation in NMD remains a clinical challenge, as indicated by the high heterogeneity regarding the optimal site and technique for the evaluation of bone quality in these patients. Although DXA is currently the diagnostic modality of choice, a consensus regarding the optimal site for BMD measurement, and the adjustment method for its obtained measurements for parameters such as age and height is needed.

## 1. Introduction

The activity of the muscle–bone interface is closely related to the normal development of the growing skeleton, while it has been also shown that bone geometry can be affected by applied muscular forces [1,2]. However, the muscle–bone interaction is altered in neuromuscular diseases (NMD), resulting in several changes in bone geometry and bone quality [3]. Patients with NMD tend to have bones of smaller diameter, while the bone density is significantly decreased in the metaphyseal area, leading to an increased fracture risk. Therefore, an accurate assessment of the fracture risk based on evaluation of the Bone Mineral Density (BMD) in patients with NMD is necessary, while it can also guide certain preventive measures to avoid insufficiency fractures in this population.

However, the results of the literature regarding the relationship between low bone quality and NMD, and the diagnostic modalities that are currently used for the evaluation of the bone quality in this population, are highly varied. This could be attributed to the fact that there are numerous entities with different pathogenetic mechanisms included in NMD, while their main pathology also affects different aspects of the muscle–bone unit such as muscles, neuromuscular junctions, peripheral nerves, and the anterior horn cells. Moreover, the population most affected by these diseases are children and adolescents, in which evaluation of BMD is problematic due to the growing skeleton, resulting in heterogeneity among the protocols used for this measurement.

Although the existing literature lacks high quality evidence regarding the evaluation of bone quality and the subsequent fracture risk in this population, various guidelines for early diagnosis and treatment have been published [4,5]. Based on the International Society for Clinical Densitometry (ISCD) guidelines, osteoporosis in children is defined as DXA BMD Z score < −2 (using age-, gender- and height-matched norms on the DXA measure), or presence of a pathologic long bone fracture/non-traumatic compressive vertebral fracture, regardless of the BMD score [4]. Although DXA remains the most widely used method to measure BMD in pathologic bone conditions, there are many controversies regarding its use in the developing skeleton, mainly regarding the optimal site and technique for measurement. Furthermore, technical difficulties due to scoliosis and alterations in hip anatomy (subluxation, luxation) in these patients do not always allow for proper measurements, while metallic implants in the two most frequent sites for DXA, the hip and the lumbar spine, also produce artifacts and are considered a contraindication [6].

The purpose of this scoping review is to investigate the current literature regarding the most widely used methods for the evaluation of bone quality in patients with BMD and to review the most recent guidelines for the assessment of the fracture risk in this population in order to provide clinicians with an updated review of the diagnostic approach for the evaluation of BMD in patients with NMD.

## 2. Methods

### 2.1. Search Protocol/Databases

A protocol was designed based on the Preferred Reporting Items for Systematic Reviews and Meta-analyses (PRISMA) guidelines in order to identify and assess studies that were evaluating BMD in patients with NMD. The PubMed and Scopus databases were queried to identify eligible studies utilizing a combination of words pertinent to BMD in NMD, such as “neuromuscular disease,” “bone mineral density,” “osteopenia” and “pediatric bone metabolism.” Following an initial screening of the titles and abstracts of the retrieved articles for eligibility, studies that were clearly irrelevant to the topic of interest were excluded. The rest of the studies underwent a full text review to assess whether they met the inclusion criteria. A data charting form was developed in order to document the variables extracted from each study. Extracted data included study characteristics, diagnosis, population age, method of BMD evaluation, body area of BMD evaluation, adjustment method of the obtained measurements, and main results of BMD evaluation. We grouped the studies by the NMD they assessed, and summarized the population, method of BMD evaluation, and body area of BMD evaluation for each group, along with the adjustment method for the obtained measurements. Two authors (GA, PM) conducted the search independently and extracted data from each study. Differences between reviewers were discussed until agreement was achieved; otherwise, disagreements were resolved by a third author (VN).

### 2.2. Selection Criteria

We considered randomized clinical trials, cohort studies (retrospective or prospective), and observational studies (case series or case reports) published in the English literature after 2007 that evaluated BMD in pediatric patients with NMD. Studies that evaluated patients > 20 years, studies not involving humans, and studies investigating bone mineral density in various pediatric conditions including NMD, but without specific data on NMD, were not included in this scoping review. The reason that only literature from after 2007 was reviewed is because the current guidelines for diagnosis of osteoporosis in children with NMD were established in 2007; thus, the review aims to evaluate the recent clinical practice after the release of these guidelines.

## 3. Results

The initial electronic search of both databases resulted in 2046 records. Following duplicate removal, 2032 records were retrieved for screening. After reviewing the abstracts/titles of these records, 1498 were considered clearly irrelevant to the topic of interest and were excluded, mainly because they evaluated different pathologies. In the following step, 534 articles were evaluated thoroughly after a full text review. After a full text review, 188 studies were excluded because they did not report any measurement of BMD, 162 studies were excluded because they did not involve humans, and 154 studies were excluded because they evaluated patients older than 20 years. Therefore, a total of 19 studies measuring BMD in pediatric patients with NMD were included in the current review. The flowchart of the reviewing process is summarized in Figure 1.

Most studies (n = 11) included patients with Duchenne Muscular Dystrophy (DMD), while four studies evaluated patients with Spinal Muscular Atrophy (SMA), one study evaluated patients with Congenital Myotonic Dystrophy (CMD), and the remaining four studies investigated BMD in patients with various NMD. In the majority of the studies (n = 17), BMD was measured in the lumbar spine. Total body BMD was measured in thirteen studies, while adjustment for Total body–Head BMD was performed in seven of them. Adjustment of BMD Z-score for parameters such as age, gender, height, and bone size highly varied among the included studies. Lastly, the fracture history was additionally documented in 11 studies.

### 3.1. Duchenne Muscular Dystrophy

Overall, 11 studies including 1099 boys addressed the evaluation of BMD in patients with DMD [7,8,9,10,11,12,13,14,15,16,17] (Table 1). BMD of the lumbar spine was measured in nine studies, BMD of the proximal femur in one study, and BMD of the lateral distal femur BMD in one study. Moreover, Söderpalm AC et al. measured BMD of the heel and forearm, while Crabtree NJ et al. used peripheral quantitative computed tomography (pQCT) to measure BMD of the distal radius [7,8]. Lastly, total body BMD was measured in ten studies, five of which used the total body/head aBMD ratio to make any comparisons.

Low BMD in children with DMD was demonstrated in all studies, especially after loss of ambulation [9], while an association between decreased muscle strength and lower BMD was shown in two studies [7,12], and an association between scoliosis and low BMD was shown in one study [17]. A correlation between lower BMD and older age was also shown in patients who were not on antiosteoporosis treatment at almost all body sites. However, this correlation was not shown in the lumbar spine in two studies by Doulgeraki et al. and Crabtree NJ et al. [8,11]. The fracture history was assessed in seven studies. It was shown that younger children with DMD tend to have greater incidence of fracture in long bones compared to older patients, in whom vertebral fractures were more common [9]. Recent clinical studies also linked lower BMD with greater fracture risk, while symptomatic vertebral fractures tend to be associated with lower lumbar spine BMD scores than asymptomatic vertebral fractures [13,15].

### 3.2. Spinal Muscular Atrophy

There were four studies evaluating BMD in 194 patients with SMA [18,19,20,21] (Table 2). BMD of the lumbar spine was measured in all these studies, while BMD of the lateral distal femur was measured in one study and total body BMD was calculated in two studies. A low BMD was found in patients with SMA in all studies, with a tendency towards lower BMD with older age [21]. Fracture history was available in three studies, and the authors of the included studies highlighted the presence of asymptomatic fragility fractures of the spine in this population [19].

### 3.3. Various Neuromuscular Disease

Four studies including a total of 767 children addressed BMD in various NMD [22,23,24,25] (Table 3). Lumbar Spine BMD was calculated in all these studies, proximal femur BMD in one study, lateral distal femur BMD in one study, and heel BMD in another study. Additionally, Total body–Head BMD adjustment was measured in one study by Söderpalm AC et al. [24].

In those studies comparing BMD among different conditions, DMD patients had lower BMD than patients with Becker Muscular Dystrophy and Cerebral Palsy [24,25]. Moreover, Kharti IA et al. evaluated BMD in 79 patients with various NMD and found that the lowest BMD scores were among patients with SMA [22]. The authors of the same study highlighted the presence of higher BMD in ambulatory children compared to non-ambulatory patients [22]. Furthermore, in a large study by Hederson et al., the authors reported lower BMD scores in the distal femur compared to the lumbar spine, while lower BMD scores in the distal were also associated with a higher risk of fracture [23].

## 4. Discussion

Τhere has been extensive research on the association between BMD and fracture risk in children with NMD, while advances in techniques and methods for BMD measurement over the last decades allow for a more accurate evaluation of the fracture risk in these patients [26]. The accumulation of bone mass during growth in patients with NMD is negatively affected by many parameters such as prematurity, low Vitamin D levels, malnutrition, antiepileptic medications, and reduced mobility [27,28,29]. Moreover, reduction of BMD begins long before loss of ambulation and the clinical focus should not be limited to non-ambulatory patients [30]. In this scoping review, we reviewed and analyzed the available data regarding the evaluation of BMD in young patients with NMD. Overall, we identified 19 studies evaluating BMD in 1983 patients with NMD, with Duchenne Muscular Dystrophy being the most widely studied disease. Dual energy X-ray absorptiometry (DXA) was the most common diagnostic modality for BMD evaluation, while the most frequent site for BMD measurement was the lumbar spine (89.4%), followed by total body BMD (68.4%).

Although DXA is the diagnostic method of choice for the evaluation of bone quality in pediatric patients, a bone biochemistry panel should be performed in every child with suspected low bone density. This panel includes serum calcium, phosphate, magnesium, creatinine, alkaline phosphatase (ALP), gamma glutamyl transferase (GGT), vitamin D, parathormone, and urinary creatinine to calcium ratio. Other laboratory studies for the evaluation of bone quality include bone turnover markers such as osteocalcin, beta crosslaps (beta-CTx), osteoprotegerin, and total s-RANKL. These markers have been used in adults for diagnosis of osteoporosis; however, their use in children and adolescents is very limited since they are elevated in the pediatric population due to the rapid bone turnover. Therefore, their specificity and predictive value is very low. Moreover, bone turnover markers are poorly correlated with lumbar BMD Z-score. However, although these markers may not be reliable indicators of low BMD, they have been shown to be decreased during therapy with bisphosphonates; therefore, they may be valuable in monitoring the response to therapy.

There is not any established protocol regarding the length of follow up or the frequency of DXA measurements, while the minimum time interval for follow-up measurements has been recommended to be no less than six months in order to avoid excessive radiation exposure. A common practice includes follow-up measurements every 6 months when on medical treatment and every 12 months when no medications are given; however, there are not any clear-cut recommendations for the follow up evaluation besides the minimum time interval of 6 months.

Management of pediatric osteoporosis is challenging. While prevention is the key, medical treatment is still debatable. Prevention of osteoporosis in children with predisposing factors such as NMD includes adequate intake of calcium and vitamin D, physical activity, treating malnutrition, correcting endocrine disorders, and avoiding osteotoxic medications such as glucocorticoids when possible. If the preventative measures are inadequate in terms of preventing low bone density on DXA or decreasing the fracture risk, bone-active agents could be prescribed. However, data regarding the efficacy, safety, and duration of treatment of these medications in the pediatric population are scarce. The anti-osteoporotic agents that are used in children include bisphosphonates, which have an anti-catabolic action, while recombinant parathyroid hormone, which has anabolic bone action, is not approved for children.

Our results indicate that physicians across the world have adapted the 2007 IDSA guidelines for evaluation of bone quality in children, which recommend DXA evaluation with posterior–anterior (PA) spine or total body minus head scans. Iolascon et al., in their review study, also noted that the most common practice for assessment of bone quality in patients with NMD follows the 2007 IDSA guidelines, with lumbar spine or total body minus head DXA being the most common diagnostic modalities [31]. However, the authors highlighted that additional DXA scans in other sites such as the proximal femur, the lateral distal femur, and the distal forearm may be useful in these patients, especially when lumbar spine or total body scans are not feasible due to certain problems such as positioning issues. These additional sites have also been reported as alternative options by the 2019 International Society of Clinical Densinometry (ISCD) pediatric guidelines [32].

DXA measures BMD in two dimensions (g/cm^2^), while the true volume of bone mass in the area under investigation (g/cm^3^) is not directly calculated. Although, in adults, this only moderately affects BMD measurements, changes in bone size and height are constant in the developing skeleton and need to be taken into consideration when evaluating BMD through DXA [5,33]. Especially for boys with DMD, height adjustments need to be made due to their short stature. Height, weight, body surface, pubertal status, bone age, and gender have been identified as important variables in BMD measurement [34].

Henderson et al., in the largest series in the literature, evaluated 619 patients with various NMD diagnoses and identified additional problems for BMD measurement and adjustment, such as inaccurate measurement of height in these children due to contractures, scoliosis, and the inability to stand [23]. To overcome this problem, they attempted to calculate lateral distal femur bone density as an alternative. Based on this technique, the distal femur was divided into three zones: Zone 1 is the metaphyseal zone, Zone 2 is the transitional zone, and Zone 3 is the shaft, with the first two being predominantly comprised of trabecular bone, like the vertebral bone, and the shaft zone being mostly comprised of cortical bone. This division is important due to the different effect of medications such as glucocorticoids on trabecular vs cortical bone, or the lack of mechanical stimulus on cortical bone. Even though this technique has been accepted as an alternative by the 2019 International Society of Clinical Densinometry (ISCD) pediatric guidelines, concerns remain due to the limited population samples and limited software analysis in the pediatric population [5,32].

Based on the 2007 ISCD guidelines, bone mass in children is recommended to be evaluated in the lumbar spine, or as total body minus head [5]. Total body DXA measurements are easier, non-invasive, reproducible, and quick, while radiation is lower and the coefficient of variation between whole body measurements is <1% in children [6]. However, certain problems, such as the inability of the child to cooperate, central obesity, and contractures in elbows, hips, and knees, may make this measurement difficult at times [35]. The skull is recommended to be excluded from the total body bone mass measurement due to its large size in children and its density; however, many studies have shown that skull contribution to total body BMD decreases with growth, which has also been observed in NMD patients, as noted by King et al. in 2014 [10].

Moreover, DXA at the lumbar spine can be problematic and may result in inaccurate measurement of the bone mass quality in these patients. In the current techniques, the measurements are adjusted based on the approximate bone volume of the vertebrae, assuming that the vertebral bodies are cylindrical, resulting in mineral apparent density -BMAD (mg/cm^3^) measurements, which are then translated into Z scores [19]. Tian et al. in their series of 292 boys with DMD noted that Lumbar spine aBMD and height for age aBMD paradoxically increased with declining motor function, despite worsening bone quality, based on whole body and distal femur DXA values [12]. The authors highlighted that special attention needs to be paid to boys with DMD because progressive loss of vertebral body height and fragility compression fractures of the spine may affect aBMD of the spine more than was expected in the past. Taking this observation into account, the ISCD Pediatric guidelines suggested the use of height Z-score adjustments for both lumbar spine BMD measurement, and the total body minus head BMD measurement [32]. Moreover, Crabtree et al. suggested that lumbar spine BMAD may not be useful in children with DMD because these are patients with not only low stature, but with additional neuromuscular imbalance, while Henderson et al. noted that the optimal site for BMD measurement may vary based on the NMD [23,36].

Another issue in BMD calculations in children is the use of the normative databases. In 2015, spine BMD Z score calculations had a difference of up to 2 SD, depending on the normative database used in each pediatric study of BMD [36]. As instructed by the ISCD Pediatric guidelines, age and gender-matched reference population of the same race, ethnicity, pubertal status, and height are the ideal normative databases [32]. In order to avoid these adjustments, which may not be accurate at times, Crabtree et al. suggested that QCT of the lumbar spine may be a better predictor of vertebral fracture in DMD patients measuring BMD in g/cm3, but larger studies are needed to evaluate this [36,37,38].

Pediatric osteoporosis is closely related to fracture risk. Many conditions related to NMD decrease bone strength and result in secondary osteoporosis, predisposing patients to fragility fractures. Common causes for secondary osteoporosis in children and adolescents with NMD include long-term use of glucocorticoids or anticonvulsants, renal disease, and endocrine disturbances such as hypogonadism. Moreover, limited physical activity and malnutrition due to low intake of proteins, vitamin D, and calcium can also affect bone mass, resulting in osteoporosis and increased fracture risk in these patients. NMD patients usually sustain lower limb fractures, as opposed to healthy children, who most commonly sustain upper extremity fractures, which further reduce their mobility and BMD [39,40,41] Moreover, these injuries can remain undiagnosed due to the low energy of the trauma or the inability of these children to communicate; therefore, radiographies are recommended every 6–24 months to prevent long periods of reduced mobility due to such undiagnosed fractures [42]. In case of back pain, whole body x-rays should be taken [43,44]. Based on the ISCD guidelines, >1 long bone fracture by the age of 10 years, or >2 long bone fractures by the age of 19 years are sufficient to establish the diagnosis of osteoporosis, without the need for DXA measurement [5]. Unfortunately, compared to adults, the threshold for fracture risk based on BMD calculations remains unknown in children with NMD and there is no study in the current literature evaluating the correlation between these two parameters.

There small number of included studies in our scoping review poses a certain limitation; therefore, our results should be interpreted with caution. The number of included studies is rather limited due to the fact that we aimed to evaluate a very specific population consisting of pediatric patients with osteoporosis and neuromuscular diseases. Moreover, we aimed to include only studies published after 2007. Therefore, the fact that we included only recent studies evaluating a very specific population resulted in the small number of included studies.

## 5. Conclusions

The ideal method for evaluation of bone quality in patients with NMD is still under investigation, as shown by the high heterogeneity in the literature regarding the diagnostic protocols that are used for BMD measurement. Although DXAis the most common diagnostic modality, several parameters, such as the optimal site of measurement, and the adjustment method for its obtained measurements, are still debatable. Moreover, BMD measurement through DXAcan be technically challenging in these non-ambulatory patients, while reference databases are lacking. In addition to these, this population undergoes constant remodeling changes due to neuromuscular imbalance, making evaluation of BMD through DXAeven more problematic. Our results indicate that the lumbar spine is the most common site for BMD measurement, followed by total body BMD, with or without excluding head BMD. A consensus regarding the optimal site for BMD measurement, and the adjustment method of its obtained measurements for parameters such as age and height is needed. The development of a bone marker representing bone modeling may also offer an alternative, and should be the focus of future research.

## Figures and Tables

**Figure 1 medicina-59-00312-f001:**
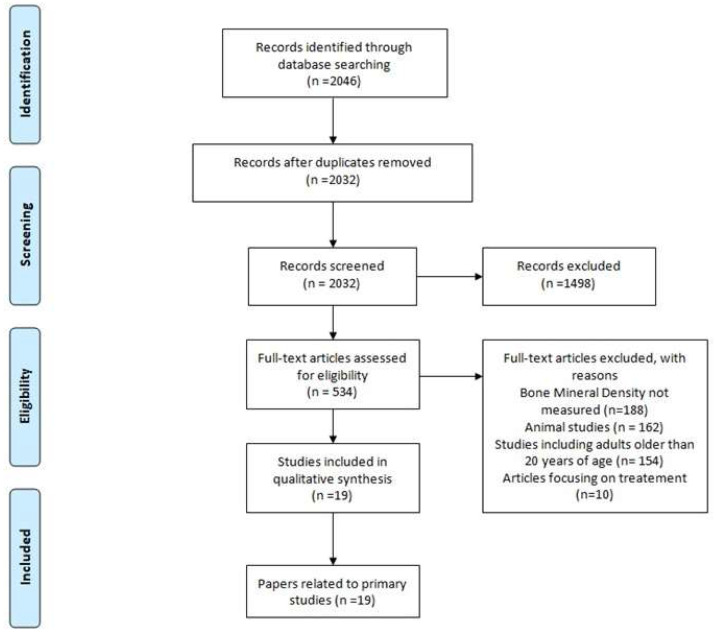
Flow chart of the literature review process.

**Table 1 medicina-59-00312-t001:** Studies evaluating bone quality in patients with Duchenne Muscular Dystrophy.

Author	Number of Patients	Age (Years)	Study	Main Findings
Söderpalm AC et al., 2007—Duchenne	24	2.3–19.7	aBMD Z-scores and BMC: total body, total body–head, lumbar spine, total hip, forearm BMAD heel Bone turnover and metabolism markers Fracture History Vignos scale	Lower aBMD at all sites compared to healthy control, more prominent with increased age Fracture rate not higher in DMD patients compared to control Decreased bone metabolism and turnover markers Muscle strength correlated with heel BMAD
Crabtree NJ et al., 2010—Duchenne	25	5.5–12	aBMD Z-scores: total body–head, lumbar spine adjusted for bone size and height BMAD calculated according to age.	Total body–head aBMD reduced with corticosteroid use Increased lumbar spine BMAD with corticosteroid use suggests that the retention of muscle function outweighs the negative effects to bone at this region
Mayo AL et al., 2012—Duchenne	39	Mean 6.6	aBMD Z score: total body–head, lumbar spine adjusted for height and age Fracture History	Height adjusted aBMD Z score stable until loss of ambulation and accrual of body fat 21% fracture history—long bones in younger children/vertebral fractures in older patients
King WM et al., 2014—Duchenne	22	5–17	aBMD Z score: total body–head adjusted for bone size and age	Low BMD Z scores Total body–Head adjusted for age which decrease even more with growth Skull contribution to Total body BMD decreases with growth
Doulgeraki et al., 2016—Duchenne	42	Mean 9.5	aBMD Z scores: total body–head, lumbar spine adjusted for age and height Bone turnover and metabolic markers	Lower aBMD at both sites compared to controls Worsening of Total body aBMD in adolescents not for Lumbar spine Bone resorption markers increased in 63% of patients
Tian C et al., 2016—Duchenne	292	5–18	aBMD Z-scores: total body–lumbar spine adjusted for age and height, lateral distal femur Fracture History Osteoporosis	Lumbar spine BMD as an isolated measure could be misleading aBMD decreased and Vertebral fractures increase with declining motor function No significant increase in osteoporosis with decline motor function
Ma J et al., 2017 —Duchenne	30	1.5–14.6	Volumetric and aBMD Z scores: lumbar spine adjusted for age and gender Fracture History	Lower Lumbar spine BMD in symptomatic vertebral fractures compared to asymptomatic 73% of patients with one fracture, 50% with vertebral fracture, 25% multiple fractures
Crabtree NJ et al., 2018—Duchenne	50	Mean 8.1	BMAD: Lumbar spine aBMD Z scores: total body–head adjusted for age, height and gender pQCT of distal radius volumetric BMD adjusted for age and gender Fracture History	aBMD Z-score Total body–Head, Total body and radius decreased significantly with age not Lumbar spine BMAD Total body and radius BMD significantly lower in children with fracture
Singh et al., 2018—Duchenne	49	Mean 14.2	aBMD Z scores: total body–lumbar spine Fracture History	aBMD Z scores decreased with increased vertebral fractures The longer the corticosteroid use the higher the risk of vertebral fracture
Joseph S et al., 2019—Duchenne	91	7.8–15	BMD Z score: total body–lumbar spine Fracture History	84% of non-vertebra fractures classified as fragility fractures No increased fracture incidence around the time of loss of ambulation
Marden JR et al., 2020—Duchenne	435	Mean 8.1 for patients on prednisone 6.5 for children with deflazacort treatment	aBMD Zscores: total body Scoliosis assessment	aBMD Zscores Whole body were the same between the two groups Lower risk of scoliosis in patients receiving deflazacort

Abbreviations: BMD, bone mineral density; BMAD, bone mineral apparent density; DMD, Duchenne muscular dystrophy; pQCT, peripheral quantitative computer tomography; BMC, bone mineral content.

**Table 2 medicina-59-00312-t002:** Studies evaluating bone quality in patients with Spinal Muscular Atrophy.

Author	Number of Patients	Age (Years)	Study	Main Findings
Poruk KE et al., 2012—SMA	47	1 month–13 years	Dietary records BMD Z-scores: lumbar spine, total body	Calcium and Magnesium intakes are independently strong predictors of increased BMD
Vai S et al., 2015—SMA	30	2–15	BMAD: lumbar spine Bone turnover and metabolic markers Fracture History	BMAD Z scores < −1.5 in 50% of patients 60% higher than normal CTx Asymptomatic vertebral fractures
Wasserman HM et al., 2017—SMA	85	1–19	aBMD Z-score: lumbar spine, total body–lateral distal femur adjusted for age, sex and race Fracture History	85% of patients had aBMD Z scores ≤ −2.0 SD Lower aBMD Z scores with increased SMA severity High prevalence of low BMD and fractures but only 12.9% met osteoporosis criteria
Baranello *G* et al., 2019—SMA	32	Mean 40	BMAD Z scores: lumbar spine adjusted for age and height Bone turnover and metabolism markers Fracture History	BMAD Z scores decreased in all patients over time Increased CTx bone resorption marker Fragility fractures more common in SMA type 2

Abbreviations: BMD, bone mineral density; BMAD, bone mineral apparent density; SMA, spinal muscular atrophy.

**Table 3 medicina-59-00312-t003:** Studies evaluating bone quality in patients with various neuromuscular diseases.

Author	Number of Patients	Age (Years)	Study	Main Findings
Khatri IA et al., 2008—Various NMD	79	4 month–18 years old	BMD Z-scores: lumbar spine	Children with SMA have the lowest BMD among other NMD patients Higher BMD in ambulatory patients with SMA compared to non-ambulatory
Henderson RC et al., 2010—Various NMD	507	6–18	aBMD Z-scores: lumbar spine, distal femur adjusted for age and gender.	Lower aBMD Z-scores in the distal femur than in the lumbar spine Consistent relationship between distal femur aBMD Z-scores and fracture in all NMD patients
Söderpalm AC et al., 2012—Various NMD	24	2.3–19.7	aBMD Zscore: lumbar spine, total body, total body–head, femoral neck, heel BMAD heel Bone turnover and metabolism markers	aBMD and heel BMAD in Duchenne Muscular Dystrophy significantly lower than age-matched Becker muscular dystrophy patients aBMD in the lower extremities strongly associated with motor function and muscle strength
Razmjou S et al., 2015—Various NMD	45	8–20	aBMD Z score: lumbar spine adjusted for age (using height age) and gender Bone turnover and metabolic markers	Duchenne patients have the lowest aBMD and Cerebral Palsy patients the highest aBMD Bone resorption markers increased in Duchenne Muscular Dystrophy

Abbreviations: BMD, bone mineral density; BMAD, bone mineral apparent density; SMA, spinal muscular atrophy; NMD, neuromuscular diseases.

## Data Availability

Data sharing not applicable.

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
