# Peer review of "A Scoping Review of the Recent Clinical Practice Regarding the Evaluation of Bone Mineral Density in Children and Adolescents with Neuromuscular Diseases"

_medicina, 2023, doi:10.3390/medicina59020312_

Round 1
Reviewer 1 Report
The manuscript of Antoniou et al. is a scoping review on bone mineral density in individuals younger than 20 yrs and with NMD. Recently bone involvement in NMD is increasingly discussed and there is a need to diagnose and monitor it properly. This manuscript is trying also to provide recommendations on the best diagnostic and follow-up methods.
In the Introduction part, the goal of the review is defined and the definition of children's osteoporosis has been provided. The methodology is sound and according to the PRISMA review standards. PRISMA workflow diagram is provided. Although the authors identified more than 2000 studies i the first run, only 20 has been selected. Results are structured into DMD, SMA and various NMDs and presented in tables. The discussion analyzed potential causes of reduced BMD in children suffering from NMD. The reference is comprehensive, however, some reviews, to which the current study could be compared, are missing. e.g. doi: 10.3389/fendo.2019.00794. The limitations of DEXA are described. Conclusions are supported by results.
Minor issues:
1. Only DEXA is discussed as a method of bone density measurements. Are there other methods available (e.g. bone turnover markers))?
2. Are there any standards for a follow-up? Are there any treatment methods? I would highlight it in the discussion.
3. I would compare the results to other similar publications in adults eg. doi: 10.3389/fendo.2019.00794
Author Response
We would like to thank the reviewer for evaluating our manuscript In the revised version of the manuscript, we tried to address adequately all points raised. Our responses to each point brought up are provided below. (Original comments in bold). Changes in the revised manuscript have been highlighted in red.
Only DEXA is discussed as a method of bone density measurements. Are there other methods available (e.g. bone turnover markers)?
Authors’ response: We agree with the reviewer that a short paragraph providing insight into other diagnostic modalities for the assessment of bone quality in pediatric patients with neuromuscular diseases would be valuable to the readers. Based on the literature, a bone biochemistry panel should be performed in every child with suspected low bone density including serum calcium, phosphate, magnesium, creatinine, alkaline phosphatase (ALP), gamma-glutamyl transferase (GGT), vitamin D, parathormone, and urinary creatinine to calcium ratio. Regarding bone turnover markers such as osteocalcin, beta cross laps (beta-CTx), osteoprotegerin, and total s-RANKL, although these markers have been used in adults for diagnosis of osteoporosis, their use in children and adolescents is very limited. These markers are elevated in the pediatric population due to the rapid bone turnover, therefore their specificity and predictive value are very low, while they are also poorly correlated with lumbar BMD Z-score. However, although they may not be reliable indicators of low BMD, bone turnover markers have been shown to be decreased during therapy with bisphosphonates, therefore they may be helpful in monitoring the response to therapy. A paragraph discussing these laboratory tests has been added in the Discussion section of the revised manuscript
Are there any standards for a follow-up? Are there any treatment methods? I would highlight it in the discussion.
Authors’ response: This information would be indeed valuable to the readers. Unfortunately, there are no clear-cut recommendations in terms of the frequency or length of DXA follow-up measurements. The recommended minimum time interval for follow-up measurements during treatment or disease progression is six months to avoid excessive exposure to radiation. Many physicians have adapted follow-up intervals of six months when on treatment and of 12 months when not treated with bone-active agents, but no official recommendations other than the stated minimum time interval of six months exist.
Management of osteoporosis in the pediatric population is challenging. While prevention is the key, medical treatment is still debatable. Prevention of osteoporosis in children with predisposing factors such as NMD includes adequate intake of calcium and vitamin D, physical activity, treating under- or overweight, correcting endocrine disturbances and avoiding osteotoxic medications, such as glucocorticoids when possible. If the preventative measures are inadequate in terms of preventing low bone density on DXA or decreasing the fracture risk, bone-active agents could be prescribed. However, data regarding the efficacy, safety, choice of agent, and duration of treatment of these medications in the pediatric population are scarce. Anti-osteoporotic agents that are mainly used in children include bisphosphonates, which have an anti-catabolic action, while recombinant parathyroid hormone which has anabolic bone action is not approved for children. Two short paragraphs discussing follow up evaluation and treatment have been added in the Discussion section of the revised manuscript
I would compare the results to other similar publications in adults eg. doi: 10.3389/fendo.2019.00794
Authors’ response: Based on the reviewer’s recommendation, a short paragraph comparing the results of our study to those of other studies such as this by Iolascon et al has been added in the Discussion section of the revised manuscript.
Reviewer 2 Report
Antoniou et al. conducted the present scoping review to investigate the recent literature regarding the assessment of BMD in children and adolescents with neuromuscular diseases, and demonstrated that DEXA is currently most commonly used for BMD evaluation though, a consensus is needed because of the high heterogeneity. While there are some concerns about the current manuscript.
Major concern:
(1)In the Selection criteria part, the authors claim that they reviewed the literature only after 2007 for that’s when the guidelines were released. In fact, the 2007 ISCD guidelines gives recommendation on the use of DXA in clinical application, for its further optimization and standardization. But it does not imply that DXA is more superior and reliable than other methods of BMD measurement. Because of the limitations of DXA and lack of research in associated fields, the gold standard for BMD evaluation in children and adolescent is not yet established. Thus, it may not be appropriate to exclude other literature based on this guidelines. It is better to give a wider time range and include a variety of BMD measurement methods for an overview of this undetermined area, which also meets the requirements of a scoping review better.
(2)As one main purpose of this review mentioned in the Introduction part, the assessment of fracture risk in the population needs to be discussed more. According to guidelines for adult osteoporosis (AACE, 2020), besides low BMD patients, patients with a history of multiple fractures, or with poor gastrointestinal and nutritional status, or with certain drug use, are also at an increased risk for future fractures. How about these situations in children and adolescents? Please discuss more.
(3)The present study synthesis data from 20 extracted studies and presents the results. Most reviews include more studies. Is this number enough for a scoping review? Please provide more evidence.
Minor concerns:
(1) The number of finally included studies, ‘19’, shown in Figure 1 is inconsistent with ‘20’ in the Result part. And it is also inconsistent with the sum of number of studies shown in Table 1, 2, and 3. Please check it.
(2) Are DEXA and DXA both abbreviations for Dual energy X-ray Absorptiometry? Please check it.
Author Response
We would like to thank the reviewer for evaluating our manuscript. In the revised version of the manuscript, we tried to address adequately all points raised. Our responses to each point brought up are provided below. (Original comments in bold). Changes in the revised manuscript have been highlighted in red.
In the Selection criteria part, the authors claim that they reviewed the literature only after 2007 for that’s when the guidelines were released. In fact, the 2007 ISCD guidelines gives recommendation on the use of DXA in clinical application, for its further optimization and standardization. But it does not imply that DXA is superior and reliable than other methods of BMD measurement. Because of the limitations of DXA and lack of research in associated fields, the gold standard for BMD evaluation in children and adolescent is not yet established. Thus, it may not be appropriate to exclude other literature based on this guidelines. It is better to give a wider time range and include a variety of BMD measurement methods for an overview of this undetermined area, which also meets the requirements of a scoping review better.
Authors’ response: We agree with the reviewer that the 2007 ISCD guidelines do not imply that DXA is the most reliable and accurate method for BMD evaluation in children and adolescents, and indeed the gold standard method for BMD evaluation in this population is yet to be established. While other diagnostic techniques such as radiogrammetry, MRI, or quantitative ultrasonography (QUS) have been proposed for the evaluation of bone health, these methods have not been tested in patients with NMD. Moreover, our aim was to investigate only the recent clinical practice regarding the evaluation of bone health in this population, which includes almost exclusively DXA, especially following the release of the IDSA guidelines in 2007. Although our study was not limited to DXA, interestingly in all studies that were published in the last 15 years, DXA was used for BMD evaluation, while in only one study peripheral quantitative computed tomography was performed (this is already mentioned in the manuscript). To be more consistent with the purpose of our study, the title of the manuscript has been amended to “A scoping review of the recent clinical practice regarding the evaluation of bone mineral density in children and adolescents with neuromuscular diseases”
As one main purpose of this review mentioned in the Introduction part, the assessment of fracture risk in the population needs to be discussed more. According to guidelines for adult osteoporosis (AACE, 2020), besides low BMD patients, patients with a history of multiple fractures, or with poor gastrointestinal and nutritional status, or with certain drug use, are also at an increased risk for future fractures. How about these situations in children and adolescents? Please discuss more.
Authors’ response: We agree with the reviewer that fracture risk is associated with many predisposing conditions to osteoporosis, such as poor nutritional status or certain drug use that are commonly seen in children or adolescents with NMD. However, all these conditions cause secondary osteoporosis which is subsequently associated with low BMD. As we state in the Discussion section, “Pediatric osteoporosis is closely related to the fracture risk”, and “The accumulation of bone mass during growth in patients with NMD is negatively affected by many parameters such as prematurity, low Vitamin D levels, malnutrition, antiepileptic medications and reduced mobility” Based on the reviewer’s recommendation, secondary osteoporosis and predisposing conditions for low BMD such as malnutrition are more extensively discussed in the Discussion section of the revised manuscript.
The present study synthesis data from 20 extracted studies presents the results. Most reviews include more studies. Is this number enough for a scoping review? Please provide more evidence.
Authors’ response: We agree with the reviewer that the number of included studies is rather limited for a scoping review. The small number of included studies in our review results from the fact that we aimed to evaluate a very specific population, consisting of not only pediatric patients with osteoporosis, but with neuromuscular diseases as well. Moreover, we aimed to include only studies that were published after 2007. Therefore, these inclusion criteria resulted in the small number of included studies. This is a certain limitation of our review which is now highlighted in the limitation paragraph of the revised manuscript.
The number of finally included studies, ‘19’, shown in Figure 1 is inconsistent with ‘20’ in the Result part. And it is also inconsistent with the sum of number of studies shown in Table 1, 2, and 3. Please check it.
Authors’ response: We thank the reviewer for noticing this error. The correct number of included studies is 19. This has been corrected across the revised manuscript.
Are DEXA and DXA both abbreviations for Dual energy X-ray Absorptiometry? Please check it.
Authors’ response: We thank the reviewer for noticing this inconsistency in the use of abbreviations in the manuscript. Indeed, both DEXA and DXA refer to Dual energy X-ray Absorptiometry. This has been corrected, and only the term DXA is now used in the revised manuscript.
Round 2
Reviewer 2 Report
Accept in present form.